# Flipped Classroom to Facilitate Deeper Learning in Veterinary Undergraduate Students: An Educational Change Pilot Study Limited to the Imaging Module Bones

**DOI:** 10.3390/ani13091540

**Published:** 2023-05-04

**Authors:** Sibylle Maria Kneissl, Alexander Tichy, Sophie Felicia Mitlacher

**Affiliations:** 1Diagnostic Imaging, Department for Companion Animals and Horses, University of Veterinary Medicine, 1210 Vienna, Austria; 2Platform Bioinformatics and Biostatistics, Department for Biomedical Services, University of Veterinary Medicine, 1210 Vienna, Austria; 3eLearning and New Media, Vicerectorate for Teaching Affairs and Clinical Veterinary Medicine, University of Veterinary Medicine, 1210 Vienna, Austria

**Keywords:** flipped classroom, veterinary medicine, education, training

## Abstract

**Simple Summary:**

All veterinary undergraduate students used an existing eLearning platform to access relevant text and selected image examples before radiology class. Only for the randomly selected students in a flipped classroom (FC) was this content amended with purposeful audio content and concrete tasks. The FC consisted of 20 students and the traditional classroom (TC) consisted of 40. The mean pre-class scores difference of students in the FC versus TC was +1.7/20 points; the mean post-class scores difference was +3/20 points. The chance of answering item 10 of the formative test (describe site of the fracture) correctly was about seven times higher for FC compared to TC learners (OR = 6.96; *p* = 0.002). The questionnaire revealed more satisfaction and greater task value in the FC compared to TC (*p* = 0.048). Higher scores, greater task value, and more positive emotions are observed in the FC compared to the TC.

**Abstract:**

In a flipped classroom, learners study at home and do the ‘homework’ in class. This approach respects the limitations of memory and allows more interaction between learners. The overall vision is self-paced activities for learners with decreased boredom and greater task value, which should facilitate deeper learning. To implement a flipped classroom, a bumpy incremental change process characterized by periods of relative stillness punctuated by the acceleration of pace was planned. All veterinary undergraduate students used an existing eLearning platform to access relevant text and selected image examples before class. Only for the randomly selected students in the flipped classroom (FC) was this content amended with purposeful audio content and concrete tasks. Further, FC learners discussed their opinions in an online class forum. To measure the educational change, a pre- and post-class formative test and a standardized questionnaire for students in the FC versus in the traditional classroom (TC) were performed. To assess engagement, students were invited to measure all learning activities, categorized into attendance, or self-study. The educational change project resulted in more commitment and less resistance from teachers. The FC consisted of 20 students, while the TC had 40. The mean pre-class scores difference between FC students and TC students was +1.7/20 points, and the mean post-class scores difference was +3/20 points. The chance of answering item 10 of the formative test (describe site of the fracture) correctly was about seven times higher for FC compared to TC learners (OR = 6.96; *p* = 0.002). The questionnaire revealed more satisfaction and greater task value in the FC compared to TC (*p* = 0.048). FC students invested 21 h into the course on average, while TC students invested 16 h. The results of this pilot agree with previous reports: A transparent process was helpful to initiate mainly positive interactions between teachers and students. Higher scores, higher chance to give the correct answer, greater task value, and more positive emotions are observed in the FC compared to the TC. Higher measures of learning time are not expected to affect exam results but indicate more engagement.

## 1. Introduction

In a flipped classroom, learners study at home and do the ‘homework’ in class. In the current face-to-face radiology training system (year 3), based on the evidence in the literature [1,2,3,4], a pilot flipped classroom called bones will affect the learning of students.

Students, as with all human beings, mainly operate depending on the social positions in which they develop and live [5]. “Engagement” is a state of vigor, dedication, and absorption [6]. Too-high demands and unfair rewards are drivers against engagement [5,6]. Veterinary students are inherently motivated to help and treat animals. This intrinsic force for action is supported by learning more about diagnostic imaging and clinical reasoning [7,8]. “Autonomy” is the right to or condition of self-government, which permits learners to take more onus for their own learning. More autonomy increases vitality, creativity, well-being, and performance [5]. Autonomous support is the didactic attempt to offer students a classroom environment that supports the need for autonomy. In contrast, control-orientated teaching prescribes the way to think, feel, or behave. Reeve et al. [9] describe the dialectic conjunction that students have with teachers and corresponding learning activities. In simple words, a motivating (not controlling) teaching style and dedicated learning activities result in student satisfaction. Such satisfaction, in turn, enhances the quality and extent of self-governed engagement. In practice, a motivating style means catching the learner’s attitude, animating inner impetus, providing answers to requests, endorsing negative emotions, relying on instructive, non-pressuring wording, and displaying patience [9,10,11].

The rationale is based on the concept of the cognitive load theory [12,13], which is built upon two commonly accepted ideas. The first is that there is a limit to how much new information the human brain can process at the same time. The second is that there are no known boundaries to how much accumulated data can be passed on at the same time [14,15,16]. Hence, the overall aim is to increase instructional opportunities that fit within the characteristics of a working memory aiming at reducing cognitive load in class [14]. The other theory this rationale is based on is the participation metaphor [17]. The participation metaphor sees learning as a social interaction where the learner constructs content meaning, beliefs, or behaviors from active participation in a community of practice. Students have different approaches to learning: surface, deep, and strategic [18]. Deep learning follows behaviors such as the intention to understand the material. Deep learning approaches are related to higher-quality outcomes, better grades, and more satisfaction [19]. Hence, designing learning activities with concrete tasks might moderate learning from surface to deep learning [20,21,22], fostered by an eLearning community of practice. The third theory is based on the concept of a zone of proximal development [23], defined as the distance between the current and the potential development level of a learner. The current level matches what a learner can do without assistance, and the proximal zone of development corresponds to what a learner can do with the guidance of an expert or in collaboration with more competent peers. Therefore, another aim of this study is to create opportunities for collaborative learning, which in turn might foster the proximal zone of development.

This project is amending the current system where an online eLearning platform (Blackboard Learn, Anthology Inc., Boca Raton, FL, USA) exists and files with selected cases are stored for students in a dedicated course to prepare for pre-class and discuss the images in class. However, they are stored as stand-alone items, and students must be eager to locate them and understand what they are expected to do. The overall vision for this project is that students are active rather than silent recipients, the learning environment is flexible and intentional, and teachers are coaches rather than traditional lecturers. At the core of the new educational design is self-paced learning, decreased boredom, and greater task value so that deep learning is facilitated. The change discussed in this project is limited to one module called bones and serves as a pilot for larger changes.

## 2. Materials and Methods

### 2.1. Approval and Study Design

All students were informed about the purpose and content of the study. Students deliberately documented their study time and submitted answers to a satisfaction questionnaire; therefore, an exemption to the requirement for ethical approval was granted by the Ethics Committee of the Medical University of Vienna. Students also knew that their personal data were unidentifiable; thus, no ethical committee approval was required. 

### 2.2. Methods for the Strategic Plan of the Educational Change Project

To implement a flipped classroom, a bumpy incremental change process characterized by periods of relative stillness and punctuated by the acceleration of pace was planned. A specific action plan using Kotter’s Eight-Stage Process for Successful Organizational Transformation [24] was developed right away and streamlined repetitively with additional ideas formed while thinking about the vision, educational gap (Appendix A), stakeholders (Appendix B), and SMART goals (Appendix C). The project was conceived as a linear model of change. Similarly, it is likely to be an open-ended, ongoing process that will have to be aligned and realigned with individual opinions or periodic questionnaires. A stakeholder analysis was performed to identify all relevant parties and potential sources of resistance. SMART goals were defined by the guiding team. An empathy map served as a tangible document [25] to establish a common ground for understanding what students say, think, do, and feel. A gap analysis was performed to understand where we are now and where we want to be. The new setting was illustrated to improve understanding in the guiding team.

### 2.3. Methods for the Flipped Classroom

When the academic year starts, a staff member of the Vicerectorate for Teaching Affairs and Clinical Veterinary Medicine divides 200 veterinary undergraduate students randomly into 10 groups. At Diagnostic Imaging, two groups (40 students) were randomly assigned to a control group and one group (20 students) to attend the flipped classroom (FC). The grouping was triggered by best matches regarding learner and teacher (S.M.K.) time slots. Further, two control groups might balance variation in students' performance. All students used an existing eLearning platform (Blackboard Learn, Anthology Inc., Boca Raton, FL, USA) for students of the FC, this content was amended with purposeful audio content and concrete tasks. Table 1 provides an overview of the study design. Further, FC learners discussed their opinions in an online class forum. To measure the educational change, a pre- and post-class formative test and a standardized questionnaire (Appendix D) (Poll Everywhere, Poll Everywhere Inc., San Francisco, CA, USA) for students in FC versus in traditional classroom (TC) were performed. To assess engagement, students were invited to measure all learning activities (time load) [26]. Time load was individually assessed using the mobile App Studo (Studo GmbH, Graz, Austria). The time load entries consisted of the time required for attendance, self-study, and writing student papers per course. Entries could be changed retrospectively for up to 106 days (correlating to the time span of the longest course). A free-text field was also available for students to submit additional comments.

### 2.4. Statistical Analysis

Data were analyzed using IBM SPSS v28. Results of pre- and post-class formative tests for FC and TC were mainly reported descriptively since specific student-IDs could not be assigned to each other. However, to measure differences in the outcome of FC versus TC in the longitudinal cohort study, two relevant items of the pre- and post-class formative tests were analyzed using chi square (χ^2^) test and summarized using the odds ratio (OR). The difference in satisfaction scores between the two groups was analyzed using the Mann–Whitney U-test. A *p*-value below 5% (*p* < 0.05) was seen as significant.

## 3. Results

### 3.1. Educational Change Project Strategy

The educational change project was introduced on 22 September 2022 at the end of a regular group meeting. The invitation to be part of a guiding team was accepted by six educators (S.M.K. included). One guiding group member (S.M.K.) participated actively in the FC teaching, while five supervised the pilot. The first guiding team meeting was on 28 September 2022. The guiding group produced an empathy map, and the opportunity was used to define roles as well as to describe the process and the anticipated product. The overall vision, the educational gap (Appendix A), stakeholder analysis (Appendix B), and SMART goals (Appendix C) were revised. It was obvious that the two senior members had supporting and critical voices reaching from no change needed to cultural change expected (from teacher- to learner-centered). The three younger team members behaved as supportive contributors. Additionally, the group reflected on the heterogeneity of learners, including learning attitudes, anticipated fears, and expectations, as well as the importance of good feedback from teachers. Hence, all tools used were useful to describe the project. Critical voices could be felt from the first moment of the educational change project. Major resistance was not evident during the meeting. We agreed on getting written feedback on what has been said, heard, and provided after the first guiding team meeting. All were assured that this is a bottom-up change project where growth hopefully equals change, and the writer’s role is providing and enabling [27]. 

Later, three persons gave written feedback: One opposing and two supporting. The critical arguments were that there is nothing new to be acknowledged, online material is existing, and the in-class session is interactive. S.M.K. reminded of supportive facts for the change project, such as more pre-class activities, including concrete interactive tasks, as well as pre- and post-class testing compared to providing access to reading material and imaging examples with limited guidance. 

The action plan helped to understand the need to form a guiding team to design the educational change process as a bottom-up change. The writer mainly expects the guiding team to quality-assess suggestions and shape the product together into a fine-tuned local asset. This agrees with the change characterized by scale [28].

### 3.2. eLearning Environment and Workshop

All veterinary undergraduate students used an existing eLearning platform. The eLearning platform is an online learning application where teachers and students can interact by posting or retrieving learning material (text, images, audio, or video), lesson plans, or selected literature. The content can be organized using learning modules or simply folders to help students to navigate through the course content. However, the online material is stored in an interdisciplinary course where course work from other disciplines is stored, too, and a student must be eager to login into the correct interdisciplinary course, navigate within the course to the discipline (Diagnostic Imaging), and then identify and select relevant material before the radiology lessons.

Only for the randomly selected FC students was this content amended with a general introduction to orientate within this module, purposeful audio content to focus on the most relevant information, and concrete tasks. The purposeful audio content included two PowerPoint files with embedded audio, one with collected statements about reading fractures, and one about reading singular bone lesions. Each PowerPoint file had 4–5 slides and focused on major and minor imaging signs (i.e., reading bone fractures and osteolytic lesions) [29]. The concrete tasks consisted of four activities (i.e., understanding imaging patterns of fractures; classifying Salter-Harris fractures; describing fractures, the normal sequence of fracture healing, and abnormal fracture healing). One example is provided in Appendix E. FC learners submitted a total of 54 comments into the online class forum to share opinions and clarify questions. Peer feedback was friendly, interactive, focused, and constructive. Sometimes students reminded each other of the information not given but overseen. In other instances, not helpful classifications, such as type of osteolysis, were correctly questioned by students. 

The workshop task for FC learners was to design the lesson. Students posed a total of 17 questions that demonstrated high interest, good preparation, the necessity to document new terminology with images, and problems that learners entering clinics have with clinical reasoning. Most learners were active. Roles of questioning, listening, or noting learners were well distributed. Questions mirrored prior understanding that needed minor or major reconstruction of knowledge. The educator (S.M.K.) guided learning by suited illustrations or verbal descriptions; she could build upon a solid socket of knowledge and skills. Overall, the observed readiness for understanding imaging patterns was higher among FC compared to TC learners. The questions of learners challenged the teacher and reduced boredom while teaching.

### 3.3. Test Results and Time Load Measures

The FC consisted of 20 students, while the TC consisted of 40. The mean pre-class scores difference between FC students and TC students was +1.7/20 points; the mean post-class scores difference was +3/20 points (Table 2). 

Odds ratios for selected test items were stated in Table 3. While the two groups did not differ in the pre-test regarding the number of correct answers to item 10 (describe site of the fracture) or 20 (describe a cortical lesion), both items show significant differences in the post-class test in favor of FC learners. The chance of answering item 10 correctly was about seven times higher compared to TC learners (OR = 6.96; *p* = 0.002). Between the pre- and post-class test results, there were no significant improvements for item 20; but a significant effect regarding item 10 was demonstrated for FC learners (OR = 12; *p* < 0.001). 

FC students invested 21 h into the course on average, while TC students invested 16 h. No student has used the available free-text field for additional comments within the App Studo (Studo GmbH).

### 3.4. Questionnaire

The questionnaire revealed more satisfaction and greater task value in the FC compared to TC (*p* = 0.048). Further, students´ perspectives were collected using an open-end question as the last item (Appendix D). Three responses from TC learners concentrated on the need to get normal image examples or improved labelling of abnormal images in their respective figure legends. Twelve responses from the FC learners focused on the improved learning environment supporting deeper learning by (a) dedicated activities, (b) online discussions among peers, and (c) teacher feedback in a non-threatening environment (formative assessment) in the workshop. No student commented about having spent more time or feeling time pressure.

## 4. Discussion

The results of this pilot agree with previous reports: Higher scores [30], higher chance to give the correct answer [30], greater task value [31], and more positive emotions [31,32] were observed in the FC compared to the TC. A similar concept has been applied before by O`Connor et al. [1]; the authors observed 10.5% higher scores, greater task value, and more positive emotions in the flipped learning class compared to the traditional learning class. Higher measures of learning time are not expected to affect exam results but indicate more engagement [33,34,35]. Further, a transparent process was helpful to initiate mainly positive interaction between teachers and students [36,37].

Regarding the overwhelming positive emotions and extra time efforts of FC learners, it is likely that FC meets the adapted values of the new generation better [38]: The learners have high expectations and ask for transparent learning conditions [39,40]. They know how to deal with educational online resources and enjoy interacting as peers. The physical campus does not disappear, but it has a different function, such as individual feedback versus traditional classroom teaching. It is advisable that teachers from different universities interact and share online learning material to manage the new faculty workload as a collaborative community.

In the following section, selected elements for autonomous supportive teaching related to the FC project are discussed through the lens of the teacher.

### 4.1. Create an Online Working Space That Includes the Formal Framework

The online space includes the latest information about the formal framework, including good examples of various radiologic bone conditions. The framework supports the “Teach Less, Learn More” idea, which aims to engage students, prepare them for this specific task, and make learning meaningful for them [8]. Likewise, responsibility for learning is given to the student, increasing autonomy and fostering the idea of running one’s own learning journey [41].

### 4.2. Create Specific Agendas with Concrete Tasks

Meaningful tasks that address multiple goals allow for control over the learning process and engage students in evaluating their own work and progress [42]. In that setting, learners are supposed to assemble outside of class and cooperate with their team affiliates to figure out authentic issues [43,44].

Especially weak students need more time to manage learning objectives appropriately [45,46,47]. To ensure enough study time, exposure to more learning opportunities is especially useful. Students may pace their learning time to their individual needs and master the needed competencies more effectively and efficiently [48].

### 4.3. Emphasize Intragroup Cooperation and Foster Positive Peer Interaction

Cooperative learning enhances motivation and learning [49,50,51,52,53]. Conditions that support cooperation are positive interdependence [50] and individual accountability [51], which promote interaction, trust, and conflict resolution. One reason for the enhanced learning is the elaborated explanations given to peers by peers. Peer-to-peer support is a promising format that can alleviate teaching pressure for faculty and offer education to students at their own cognitive level in a safe educational environment [54,55].

By creating a peer group, learning opportunities are multiplied, and social support is increased, thus enhancing social competence, self-confidence, self-esteem, and resilience [50]. Peer coaching is also an encouraging format for social development when established within a non-evaluative environment. The overall result can be more resilience to deal with unexpected occurrences during a learning task [56].

### 4.4. Improved Feedback to Learners

Traditionally, an undergraduate veterinary student can be physically present and hardly active and will pass the radiology course when having attempted to read three to four radiologic cases under supporting teacher guidance. Hence, there is an active and individual dialogue between student and teacher in the TC, but students feel as though they are in a constant (threatening) summative assessment. The discussion boards on the eLearning platform and the face-to-face workshop implemented in the FC improved the overall feedback loop between learners and teachers by providing meaningful and good quality oral or written feedback in a non-threatening formative assessment. The power of dedicated and meaningful feedback has been demonstrated to raise self-esteem, clarify goals, and empower students for self-directed learning [57,58,59,60,61].

As stated above, one limitation of this study was the resistance of an individual teacher to change. Possibly, this educator is worried about losing resources, which is supported by the conservation of resources theory [62], or she could not grasp the chance of change as a learning opportunity [63]. Future studies may explore reasons for resistance to change and consider conditions needed for better cooperation, such as positive interdependence [49], appropriate social skills [64], resilience [65], or improved appreciative inquiry [66]. In the present case, the planned changes are meant mainly to enrich a program through interactive tasks, peer mentoring, and improved feedback that will increase and eventually ignite self-determined motivation and autonomy. Although this changing process is within the institutional logic [67] that describes itself as learner-centered [68], opinions about teaching methods or teachers being explicitly learner-centered vary. Hence, this educational change project is a further step in the same direction and guides team members to have different opinions about the necessity of change.

Another limitation of this study was that the default settings of the test software used are limited to group observations and do not allow individual tracking. It is likely that an FC depends on student performance, meaning that higher-level learners students outperform compared to lower-level learners [69]. In future studies, we will observe students as individuals.

## 5. Conclusions

The key-project aims are less cognitive load and deeper learning by implementing an FC for veterinary undergraduate students in veterinary radiology. The project was successful in observing engagement and better grades for FC students. The bottom-up bumpy incremental change resulted in more commitment and less resistance from teachers. The transparent process was helpful to initiate mainly positive interaction. This agrees with the idea of distributional leadership [70], where transparent rules and roles are an effective start [20,36,37,71].

In a subsequent project, the module bones will be amended with the module joints and thorax. This will allow the team of teachers to agree on learning outcomes, select relevant teaching material, and design corresponding tasks. In this phase, it will be essential to allow tracking of individual learners to get longitudinal information of and measure anticipated differences between learners attending TC and FC or different students’ levels. Further, it will be necessary to learn more about the veterinary students’ and educators’ perspectives regarding their needs. Debatably, these studies are needed to explore if all undergraduate veterinary students study before class and master the shift in the learning responsibility.

## Figures and Tables

**Table 1 animals-13-01540-t001:** Study design.

Chronology	Flipped (Interventional) Group	All students (or Traditional Group When Explicitly Mentioned) Participating in This Study
Before class	S.M.K. prepared a dedicated bone module using 2 PowerPoint files with embedded audio and 4 concrete activities. Further, students were provided the outline of this study. Students watched material and posted opinions in the corresponding discussion boards.	PowerPoint files including selected images with descriptions are provided as stand-alone files. Further, students have access to a DICOM image platform to enlarge or contrast selected image examples. Pre-class test Access learning time
In class	Workshop (2 h): Students pose their questions in shape of formative assessment.	Traditional Group only (4 h): face-to-face learner and teacher dialogue (mainly in the shape of summative assessment)
After class		Post-class test Questionnaire

**Table 2 animals-13-01540-t002:** Results comparing the intervention to the control group.

Averaged Per Student	Flipped (Intervention) Group	Traditional (Control) Group
Pre-Test [averaged points out of 20]	13.1	11.4
Post-Test [averaged points out of 20]	15.5	12.5
Time load [averaged hours]	20.6	16.4

**Table 3 animals-13-01540-t003:** Odds ratios of item 10 and 20 comparing FC to TC test results.

Odds Ratio	Pre-Class Test	Post-Class Test
FC versus TC result (item 10)	1.14	6.96 *
FC versus TC result (item 20)	2.7	4.5 *

* significant improvement in favor of FC learners.

## Data Availability

The datasets used and analysed during the current study are available from the corresponding author on reasonable request.

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
