# Peer review of "Flipped Classroom to Facilitate Deeper Learning in Veterinary Undergraduate Students: An Educational Change Pilot Study Limited to the Imaging Module Bones"

_animals, 2023, doi:10.3390/ani13091540_

Round 1

Reviewer 1 Report

The manuscript is embedded in an expanding field of research and teaching aimed at renewing teaching methodologies and improving learning in higher education. Active teaching approaches such as the flipped classroom are linked to better quality student learning outcomes across health care disciplines, with the potential to support students' preparedness for practice. In this regard, the results obtained confirm the benefit of using alternative teaching tools that promote the active stimulation of the student in the learning phase. That said, a few suggestions for timely changes to the text are provided below.

Simple summary

Line 16 please provide TC acronym extension.

Introduction

Line 51 the sentence is not clear please rephrase and expand the content

Line 56-58 please rephrase the sentence; the content is unclear.

Line 63 please expand the content of the sentence. Does “teacher control” refer to the teacher-centered learning approach?

Materials and methods

Why were two control groups included in the study? Why wasn't an equal number of students provided instead? Please specify the underlying rationale for creating the study populations.

Lines 126-130. Since the study was conducted on 60 students (40 placed in two control groups and 20 in FC group) is the mention to the initial 200 students necessary and relevant?

Line 132 Please provide details and examples on “purposeful audios and concrete tasks” implemented

Line 136-137 timely description of how students measured their learning activities and any scores used would help improve understanding of this aspect

Line 174 Is the eLearning platform that students can access assimilated to a database of consultable sources and materials or does it also contain interactive material? Please specify if possible

Discussion

if deemed useful, the outcomes of the study could be compared with similar published studies , following are some suggestions:

Sukut, S. L., Mayer, M. N., D’Eon, M. F., Burbridge, B. E., & Waldner, C. L. (2021). Comparing two resources used to teach pulmonary patterns for a flipped veterinary radiology course. Journal of veterinary medical education, 48(2), 211-216.

Yang, C., Yang, X., Yang, H., & Fan, Y. (2020). Flipped classroom combined with human anatomy web-based learning system shows promising effects in anatomy education. Medicine, 99(46).

Decloedt, A., Franco, D., Martlé, V., Baert, A., Verwulgen, A., & Valcke, M. (2021). Development of surgical competence in veterinary students using a flipped classroom approach. Journal of Veterinary Medical Education, 48(3), 281-288.

Line 213 Please specify which previous reports are being referred to

Author Response

Comments to Reviewer 1
The manuscript is embedded in an expanding field of research and teaching aimed at renewing teaching methodologies and improving learning in higher education. Active teaching approaches such as the flipped classroom are linked to better quality student learning outcomes across health care disciplines, with the potential to support students' preparedness for practice. In this regard, the results obtained confirm the benefit of using alternative teaching tools that promote the active stimulation of the student in the learning phase. That said, a few suggestions for timely changes to the text are provided below.
Author Comment (AC): Thank you for this kind feedback; we appreciate the work and effort you have made as our reviewer and hope that our revision meets your expectations.
Simple summary
Line 16 please provide TC acronym extension.
AC: I am sorry that we missed this, now the acronym has been provided.
Introduction
Line 51 the sentence is not clear please rephrase and expand the content
AC: I have deleted this sentence in the introduction and added the problem of suitable feedback in the discussion in chapter 4.5 as following:
4.4. Improved feedback to learners
Traditionally, an undergraduate veterinary student can be physically present and hardly active and will pass the radiology course when having attempted to read 3 to 4 radiologic cases under supporting teacher guidance. Hence, there is an active and individual dialogue between student and teacher in the TC, but students feel to be in a constant (threatening) summative assessment. The discussion boards on the eLearning platform and the face-to-face workshop, implemented in the FC, improved the overall feedback loop between learners and teacher by providing meaningful and good quality oral or written feedback in a non-threatening formative assessment. The power of dedicated and meaningful feedback has been demonstrated to raise self-esteem, clarify goals, and empower students for self-directed learning (AuthorUnknown, 2011; Barton et al., 2016; Boud & Walker, 1990; McKimm, 2009; Orsmond & Merry, 2013).

Line 56-58 please rephrase the sentence; the content is unclear.
AC: I do hope that you were pointing at the sentence *Because the main interests of veterinary medicine students lie in helping and treating animals, most of these students have an intrinsic motivation to learn more about diagnostic imaging, respectively clinical reasoning. The sentenced was revised to: * Veterinary students are inherently motivated to help and treat animals. This intrinsic force for action is supported by learning more about diagnostic imaging, respectively clinical reasoning (Klevin et al., 2013; Linn et al., 2012).* I hope this clarifies the meaning.
Line 63 please expand the content of the sentence. Does “teacher control” refer to the teacher-centered learning approach?
AC: No. Reeve (2016) contrasts autonomous (motivating) and control-orientated (less motivating) teaching when developing autonomous learners, the later resulting in decrease of autonomy, or less satisfaction. High quality learning occurs optimally in context were learners find the basic psychological needs for active self-development. We have revised teacher control to control-orientated teaching. Thank you for pointing this out.
Materials and methods
Why were two control groups included in the study? Why wasn't an equal number of students provided instead? Please specify the underlying rationale for creating the study populations.
AC: Prospectively (and as stated in Chapter 2.3.) no other rationale for grouping but best matches for teacher (S.K.) and students time slots was evident. Further two control groups might balance variation in students´ performance. This last sentence was included into Chapter 2.3.
Lines 126-130. Since the study was conducted on 60 students (40 placed in two control groups and 20 in FC group) is the mention to the initial 200 students necessary and relevant?
AC: The detailed description of the random sampling method was requested by the editor and should be included to facilitate transparent and open science.
Line 132 Please provide details and examples on “purposeful audios and concrete tasks” implemented 
AC: Details on purposeful audios and concrete tasks were added into chapter 3.2. The included statements are: The purposeful audios were 2 PowerPoint files with embedded audios, one with collected statements about reading fractures and one about reading singular bone lesions. Each PowerPoint file had 4-5 slides and focused on major and minor imaging signs (i.e., read-ing bone fractures and osteolytic lesions) (29). The concrete tasks consisted of 4 activities (i.e., understanding imaging patterns of fractures; classifying Salter-Harris fractures; de-scribing fractures, the normal sequence of fracture healing and abnormal fracture healing). One example is provided as Appendix E.
Line 136-137 timely description of how students measured their learning activities and any scores used would help improve understanding of this aspect
AC: Thank you for mentioning this. The following dedicated information regarding the citation (Kneissl et al., 2022) has been included into the text: Time load was individually assessed using the mobile App Studo (Studo GmbH). The time load entries consisted of the time required for attendance, self-study, and writing student papers per course. Entries could be changed retrospectively for up to 106 days (correlating to the time span of the longest course). A free-text field was also available for students to submit additional comments.
Line 174 Is the eLearning platform that students can access assimilated to a database of consultable sources and materials, or does it also contain interactive material? Please specify if possible
AC: You are so right mentioning this. Activities for the FC were interactive, material for TC corresponded to stored pdf files and access to a DCM-based imaging archive. The text has been revised such as: S.M.K. reminded of supportive facts for the change project such as more pre-class activities including concrete interactive tasks as well as pre- and post-class testing compared to providing access to reading material and imaging examples with limited guiding.
Discussion
if deemed useful, the outcomes of the study could be compared with similar published studies, following are some suggestions: 
Sukut, S. L., Mayer, M. N., D’Eon, M. F., Burbridge, B. E., & Waldner, C. L. (2021). Comparing two resources used to teach pulmonary patterns for a flipped veterinary radiology course. Journal of veterinary medical education, 48(2), 211-216.
Yang, C., Yang, X., Yang, H., & Fan, Y. (2020). Flipped classroom combined with human anatomy web-based learning system shows promising effects in anatomy education. Medicine, 99(46).
Decloedt, A., Franco, D., Martlé, V., Baert, A., Verwulgen, A., & Valcke, M. (2021). Development of surgical competence in veterinary students using a flipped classroom approach. Journal of Veterinary Medical Education, 48(3), 281-288.
AC: Thank you for adding some useful citations; these were embedded as # 30-32. Another additional corresponding citation is Oudbier et al. (2022).
Line 213 Please specify which previous reports are being referred to
AC: Sadly, Line 213 is not part of the discussion, so I cannot comment this feedback. 

Author Unknown (2011). 7 steps to effective feedback. Plymouth University. https://www.plymouth.ac.uk/uploads/production/document/path/2/2394/7_steps_effective_feedback_2011__3_.pdf 
Barton, K. L., Schofield, S. J., McAleer, S., & Ajjawi, R. (2016). Translating evidence-based guidelines to improve feedback practices: the interACT case study. BMC medical education, 16(1). https://www.ncbi.nlm.nih.gov/pmc/articles/PMC4748473/ 
Boud, D., & Walker, D. (1990). Making the most of experience. Studies in Continuing Education, 12, 61-80. 
Klevin, T., Jude, C., & Tan, C. (2013). Engaging our learners: teach less, learn more. In J. Larkley & V. Maynhard (Eds.), Innovation in Education (pp. 153-177). National Library Board. 
Kneissl, S., Tomiska, T., & Rehage, J. (2022). Measuring Time Load Using a Mobile Application to Monitor Curriculum Workload. Journal of Veterinary Medical Education, e20210127. 
Linn, A., Khaw, C., Kildea, H., & Tonkin, A. (2012). Clinical reasoning - A guide to improving teaching and practice. Aust Fam Physician, 41(1-2), 18-20. https://www.racgp.org.au/afp/201201/45593 
McKimm, J. (2009). Giving effective feedback. British Journal of Hospital Medicine, 70(3), 158-161. 
Orsmond, P., & Merry, S. (2013). The importance of self-assessment in students’ use of tutors’ feedback: a qualitative study of high and non-high achieving biology undergraduates. Assessment & Evaluation in Higher Education, 38(6), 737-753. https://doi.org/10.1080/02602938.2012.697868 
Oudbier, J., Spaai, G., Timmermans, K., & Boerboom, T. (2022). Enhancing the effectiveness of flipped classroom in health science education: a state-of-the-art review. BMC Med Educ, 22(1), 34. https://doi.org/10.1186/s12909-021-03052-5 
Reeve, J. (2016). Autonomy-Supportive Teaching: What It Is, How to Do It. In L. Woon Chia, J. Keng, & R. Ryan (Eds.), Building Autonomous Learners. Springer. 

Reviewer 2 Report

GENERAL COMMENTS

This is an interesting manuscript and easy to read.

Hypothesis statement is clear.

Deployed methodology is overall clear. Some clarifications would be necessary.

Results session needs some improvement.

Discussion and conclusions could also be improved.

 SPECIFIC COMMENTS

1.       Introduction

·       Line 95: I do not exactly understand how the platform is established and what students should do to locate the appropriate material. Perhaps you could include a short description of the platform. It may be appropriate to include such description in materials and methods.

2.       Methodology

The methodology is simple and overall clear.

Line 130: What is the meaning of the author’s initials (S.K.) here?  Is it a typo mistake or does it have a meaning – if the second please explain.

3.       Results

·       Line 150: I would suggest the following amendment “ … guiding team was accepted by six educators…”

·       Did all 6 educators participated actively in flipped method teaching or some of them were only supervisors of the pilot?

·       What were the clear tasks these 6 persons were committed to according to the project plan? This description is missing.  

·       Were the educators assessed for their satisfaction? Were they asked for feedback and recommendations after this pilot? If so, please present this information and perhaps present if there was any change in their initial opinion about the project – if not, the lack of assessment of the educators’ satisfaction/opinion may be noted as a limitation.

·       What I also miss is a short description of the TC. I understand that students of the TC have also access to the e-learning platform. It would be useful to describe both educational models – FC and TC – and highlight the different approaches. Also describe a bit better the e-learning platform and its use by the students (see also comment above)

·       Lines 196 – 200: The satisfaction of educators should be presented as well, if such information exist.

4.       Discussion

·        Discussion has to be adapted depending on potential new information that could be provided by the authors in reply to the above comments. That could include for example

o   A comparison between FC and TC outcomes form educators’ and students’ perspectives. E.g. lines 235-237 discuss how (if so) FC support better the weak students than TC considering that all student have access to the platform?

o   Line 244: I see the benefits of peer-to-peer support among students, however I would like to see some discussion about potential burden on the “good students” that take over the task of teaching their fellows. They can certainly “alleviate teaching pressure for faculty” but veterinary students have also a very heavy programme. This would be an important point to look into in the future if you implement individual students observation.

o   On lines 256-257 you refer to the resistance of individual teachers to change. What are the reasons for this resistance? Could you please discuss about is, e.g. does it have to do with specific arguments? Please see also above on the need for better description of the instructors’ profile, tasks, assessment of their satisfaction and feedback on this approach.

·       Lines 264-268: Depending on the response to all above comments, if information and feedback  on educators cannot be provided, it should be noted as a limitation of the study. I think we must consider both students’ and teachers’ perspectives/needs/ feedback when we are assessing teaching methods as both of them are core contributors to the success of any educational model.  I also suggest that this paragraph is clearly separate from the previous one (session 4.4)

5.       Conclusion

Depending on the integration of all above comments, conclusions should be adapted respectively.

Thank you for your consideration. I hope you will find these suggestions useful.

Kind regards

Author Response

Comments to Reviewer 2

GENERAL COMMENTS
This is an interesting manuscript and easy to read.
Hypothesis statement is clear.
Deployed methodology is overall clear. Some clarifications would be necessary.
Results session needs some improvement.
Discussion and conclusions could also be improved.
Author Comment (AC): Thank you for this kind feedback; we appreciate the work and effort you have made as our reviewer and hope that our revision meets your expectations.
SPECIFIC COMMENTS
1.       Introduction
Line 95: I do not exactly understand how the platform is established and what students should do to locate the appropriate material. Perhaps you could include a short description of the platform. It may be appropriate to include such description in materials and methods.
AC: Thank you for addressing this. The following description has been included into Chapter 3.2.: The eLearning platform is an online learning application where teachers and students can interact by posting or retrieving learning material (text, images, audio, or video), lesson plans, or selected literature. The content can be organized using learning modules or simply folders to help students to navigate trough the course content.
I do hope this is clearer now.

2.       Methodology
The methodology is simple and overall clear.
Line 130: What is the meaning of the author’s initials (S.K.) here?  Is it a typo mistake or does it have a meaning – if the second please explain.
AC: Thank you for mentioning this. S.K. was revised to S.M.K.

3.       Results
·  Line 150: I would suggest the following amendment “ … guiding team was accepted by six educators…”
AC: Revised.
·  Did all 6 educators participated actively in flipped method teaching or some of them were only supervisors of the pilot? 
AC: Thank you for pointing this out. The following text has been added: One guiding group member
(S.M.K.) participated actively in the FC teaching, five supervised the pilot.   
·  What were the clear tasks these 6 persons were committed to according to the project plan? This description is missing.  
AC: An empathy map was produced as reported in L160. The following text has been included: The overall vision, the educational gap (Appendix A), stakeholder analysis (Appendix B), and SMART goals (Appendix C) were revised.
·  Were the educators assessed for their satisfaction? Were they asked for feedback and  recommendations after this pilot? If so, please present this information and perhaps present if there was any change in their initial opinion about the project – if not, the lack of assessment of the educators’ satisfaction/opinion may be noted as a limitation.
AC: In this pilot there was no assessment of educators for their satisfaction because there was only one trainer (the author S.M.K.) involved into the FC training. It will make sense to get feedback from trainers when being involved into the FC training in winter 2023. Hence, thank you for raising this issue but I do not think that any information is missing.  
·  What I also miss is a short description of the TC. I understand that students of the TC have also access to the e-learning platform. It would be useful to describe both educational models – FC and TC – and highlight the different approaches. Also describe a bit better the e-learning platform and its use by the students (see also comment above)
AC: I agree that the study design could be presented more transparent, hence I included Table 1. Study design was added. I hope that this clarifies the differences between the interventional and the control group.
·  Lines 196 – 200: The satisfaction of educators should be presented as well, if such information exist.
AC: Good point. This perspective of the involved educator has been with the following statements: Most learners were active. Roles of questioning, listening, or noting learners were well dis-tributed. Questions mirrored prior understanding that needed minor or major reconstruc-tion of knowledge. The educator (S.M.K.) guided learning by suited illustrations or verbal descriptions; she could build upon a solid socket of knowledge and skills.

4.       Discussion
·  Discussion has to be adapted depending on potential new information that could be provided by the authors in reply to the above comments. 
AC: Your feedback guided me to improve the description of the study design, respectively differences of the groups (see Table 1 and Appendix E), as well as providing more details regarding the eLearning platform. Thank you for this valuable opportunity! Regarding number of educators, there must have been a misunderstanding caused by my misleading abbreviation (not always including my middle name). S.M.K. (me) was the only educator involved in this pilot. Hence, I think that the discussion through the lense of a teacher´s perspective is still useful to support readers including more traditional teachers to switch to a learner centered autonomous supportive teaching style. The student´s perspectives were collected using an open-end question included into the questionaire. This information was added to the manuscript at all relevant instances (Chapter 3.3., Appendix D, second paragraph in discussion).

That could include for example
o   A comparison between FC and TC outcomes form educators’ and students’ perspectives. E.g. lines 235-237 discuss how (if so) FC support better the weak students than TC considering that all student have access to the platform? 
AC: I included more literature regarding the outcome: Higher scores (Sukut et al., 2021), higher chance to give the correct answer (Sukut et al., 2021), greater task value (Yang et al., 2020) and more positive emotions (Decloedt et al., 2021; Yang et al., 2020) were observed in the FC compared to the TC.). As mentioned above, I added the perspectives from students (Chapter 3.3., Appendix D, second paragraph in discussion, Chapter 4.4).
o   Line 244: I see the benefits of peer-to-peer support among students, however I would like to see some discussion about potential burden on the “good students” that take over the task of teaching their fellows. They can certainly “alleviate teaching pressure for faculty” but veterinary students have also a very heavy programme. This would be an important point to look into in the future if you implement individual students observation.
AC: This is a very good point. We have enough time to plan students observations for the next phase of this educational change project. Regarding this manuscript I embedded this idea into future studies such as: Further it will be necessary to learn more about the veterinary students´ and educators´ perspective regarding their needs. Debateably, these studies are needed to explore if all undergraduate veterinary students study before class and master the shift in the learning responsibility.
o   On lines 256-257 you refer to the resistance of individual teachers to change. What are the reasons for this resistance? Could you please discuss about is, e.g. does it have to do with specific arguments? Please see also above on the need for better description of the instructors’ profile, tasks, assessment of their satisfaction and feedback on this approach.
AC: Please see notes above (S.M.K was the only educator involved in teaching during this pilot study). I can only speculate about reasons for resistance but attempted to include the following lines: Possibly, this educator is worrying to lose ressources which is supported by the conserva-tion on ressources theory (62), or she could not grasp the chance of change as a learning opportunity (63). Future studies may explore reasons for resistence to change and consider conditions needed for better cooperation such as positive interdependence (49), appropriate social skills (64), resilience (65), or improved appreciative inquiry (66).
·       Lines 264-268: Depending on the response to all above comments, if information and feedback on educators cannot be provided, it should be noted as a limitation of the study. I think we must consider both students’ and teachers’ perspectives/needs/ feedback when we are assessing teaching methods as both are core contributors to the success of any educational model.  
AC: Please see notes above (S.M.K was the only educator involved in teaching during this pilot study) regarding your comment to include feedback from educators. Therefore, I embedded this idea into future studies such as: Further it will be necessary to learn more about the veterinary students´ and educators´ perspective regarding their needs. Debateably, these studies are needed to explore if all undergraduate veterinary students study before class and master the shift in the learning responsibility.
I also suggest that this paragraph is clearly separate from the previous one (session 4.4)
AC: I hope you were pointing at the paragraph regarding emotions (Another important issue to address is emotions. The interplay of emotions and exhaustion can result either in mental health or in burnout (48). It is important that students are not dislocated from their feelings and that they learn to display appropriate emotion (49)). I agree that these lines clearly separate and therefore deleted them.

5.       Conclusion 
Depending on the integration of all above comments, conclusions should be adapted respectively. Thank you for your consideration. I hope you will find these suggestions useful.
Adapt conclusion 
AC: Please see notes above (S.M.K was the only educator involved in teaching during this pilot study). Hence this information was clarified. Further it will be necessary to learn more about the veterinary students´ and educators´ perspective regarding their needs. Debateably, these studies are needed to explore if all undergraduate veterinary students study before class and master the shift in the learning responsibility.
I do hope this minor revision meets your expectations. I am very grateful for your helpful efforts.

Decloedt, A., Franco, D., Martlé, V., Baert, A., Verwulgen, A., & Valcke, M. (2021). Development of Surgical Competence in Veterinary Students Using a Flipped Classroom Approach. J Vet Med Educ, 48(3), 281-288. https://doi.org/10.3138/jvme.2019-0060 
Sukut, S. L., Mayer, M. N., D'Eon, M. F., Burbridge, B. E., & Waldner, C. L. (2021). Comparing Two Resources Used to Teach Pulmonary Patterns for a Flipped Veterinary Radiology Course. J Vet Med Educ, 48(2), 211-216. https://doi.org/10.3138/jvme.2019-0049 
Yang, C., Yang, X., Yang, H., & Fan, Y. (2020). Flipped classroom combined with human anatomy web-based learning system shows promising effects in anatomy education. Medicine (Baltimore), 99(46), e23096. https://doi.org/10.1097/md.0000000000023096 
